# Deciphering climate-induced displacement in Somalia: A remote sensing perspective

Rahman Momeni[1], Tuba Bircan [2]*, Robert King[1], Eloy Zafra Santos[1]

1 GMV, Harwell, Oxfordshire, United Kingdom, 2 Dept. of Sociology, Vrije Universiteit Brussel, Brussels, Belgium

* tuba.bircan@vub.be

## Abstract

Rapid climate changes bear significant consequences on various aspects of our lives, notably by deteriorating living conditions in certain areas to such extent that inhabitants have no choice but flee. Despite recognition of this issue, the dynamics of the relationship between the environmental factors and the human mobility have yet to be thoroughly investigated. This study aims to explore the application of advanced remote sensing analytics for developing detailed climate indicators at a micro (district) level, and to examine the relationship between climate factors and internally displaced persons. After detailing our data sources and the analytics employed for indicator development, we discuss various types of events and their repercussions. Our findings corroborate that slow-onset and rapid-onset climate events differently impact society, and the responses hinge on the urgency precipitated by the detrimental aftermath of the extreme weather event and, most crucially, on people's capabilities. We also underscore the importance of data quality and availability for the socio-economic indicators to enhance future studies, given the intertwined associations between climate change, economic deprivation, and violent conflict.

**Data Availability Statement:** All data are publicly available. To be more specific, - The aggregated and anonymised tabular data used for this paper is available under the link: https://github.com/bircantuba/HMB.git - The OSF data (for the remote

## Introduction

Over the past decade, we have witnessed numerous shocks across the globe - whether political, economic, societal, or environmental - triggering humanitarian crises and prompting millions to flee or be displaced. Political instability in the Middle East and economic downturns in Latin America since 2015, for instance, have dominated international discourse as mass displacement ensued, with individuals initially seeking refuge in neighbouring countries before looking further afield [1–5]. Such crises, and the resultant emigration they provoke, have been extensively documented and rigorously studied.

As of February 2022, global attention has turned towards Ukraine as it confronts its own displacement struggle. Whilst these sudden - albeit not unexpected - events typically attract significant attention due to the immediate response required from host countries, numerous nations, particularly in Asia and Africa, have been contending with the severe impacts of chronic or acute environmental disasters for years. When compounded with enduring challenges like poverty, famine, armed violence, and political instability, these environmental

sensing analysis) can be found under the link:
https://osf.io/kg9zd/

**Funding:** MR, BT, KR This research is supported by the European Commission through the Horizon2020 European project: "HumMingBird – Enhanced migration measures from a multidimensional perspective" (GA: 870661). https://research-and-innovation.ec.europa.eu/funding/funding-opportunities/funding-programmes-and-open-calls/horizon-europe_en The funders had no role in study design, data collection and analysis, decision to publish, or preparation of the manuscript.

**Competing interests:** The authors have declared that no competing interests exist.

factors' societal effects amplify, forcing people to relocate. As a result, these individuals first experience displacement within their own countries before eventually seeking refuge elsewhere.

Indeed, the global population of internally displaced persons (IDPs) has steadily increased, with the 2021 Internal Displacement Index Report indicating that by the end of 2020, the number of IDPs worldwide had surpassed 55 million, double the number of refugees [6]. By developing a nuanced understanding of the root causes behind irregular migration and forced displacement, we can better anticipate patterns of human mobility. More importantly, we can enhance protection measures for migrants and refugees, providing them with the safety and support they need in these challenging circumstances.

The field of climate-induced migration has seen significant advances in contemporary predictions compared to earlier efforts, spurred by growing interest in studying environmental dynamics and mobility decisions [7–12]. Harnessing innovative data sources and cutting-edge analytics has enormous potential for data collection, consequently enhancing our understanding of human mobility [13, 14]. This improved knowledge can be instrumental in the development of predictive models [15, 16], understanding slow-onset processes of climate change [17], and advancing socio-economic indicator development [18].

Particularly, remote sensing technologies offer unique capacities to study the relationship between the climate change and human mobility. These technologies provide critical environmental data at varying scales, elucidating the impacts of climatic indicators, such as droughts and floods, on population movements [19]. Satellite-derived data offer insights into climatic trends, including rainfall variability and temperature anomalies, alongside visualising land use changes and environmental degradation [20, 21]. The integration of these datasets with socio-economic variables can reveal patterns and hotspots of climate-induced migrations [22]. Moreover, remote sensing data, coupled with machine learning algorithms, can be used to predict future migration patterns under various climate scenarios [23]. With this study, we will assess how to maximise the potential of these technologies in understanding the intricate interplay of climate change and human displacement.

Accordingly, the primary objective of this paper is to scrutinise the relationship between climatic factors and IDPs within the context of Somalia from 2016 to 2019. Our approach employs advanced remote sensing analytics to construct sophisticated climate indicators at a granular, district level.

The structure of this article unfolds as follows: We begin with a comprehensive examination of the case of Somalia, elucidating both the environmental conditions and the historical narrative of migration. Subsequently, we delineate our data sources and the methodological framework used for creating climate indicators. In the section analysing our results, we discuss our investigation and visualise the indicators through maps to elucidate the relationship between environmental factors and IDPs. In the conclusion, we summarise our findings and offer insights and recommendations aimed at enhancing current practices in this research domain.

## Environmental change and displacement in Somalia

Our study focuses on Somalia due to its vulnerability to extreme climatic events, which have triggered considerable internal migration. The nation's fragility and susceptibility stem from a combination of enduring conflicts, protracted epidemic outbreaks, erratic climatic conditions and shocks adversely affecting agriculture, livestock, and rural livelihoods, which are fundamental for food security. Simultaneously, weak social protection systems further exacerbate the situation. The World Bank classifies Somalia as one of the world's most impoverished countries, with approximately 70% of its population living below the poverty line, subsisting

on less than $2 per day. Furthermore, the United Nations High Commissioner for Refugees (UNHCR) places Somalia seventh globally in terms of displaced populations.

Since 1990, Somalia has endured over 30 extreme weather events, encompassing 12 droughts and 19 floods, triggering two significant famines in 1991–92 and 2011. These disasters led to more than 3.2 million new displacements, implying the number of movements rather than the number of individuals displaced. Given that over 80% of Somalia's territory features arid and semi-arid climates, and rainfall exhibits high spatial and temporal variability, these adverse environmental impacts and inadequate natural resource management negatively affect the nation's economy. Approximately 70% of Somalis are reliant on climate-sensitive agricultural and pastoral activities.

The recurring interplay of drought and floods undermines food security and access to basic services, heightening vulnerability and prompting displacement at varying scales. UNHCR data reveals that since 2016, over three million Somalis have been displaced due to recurrent extreme droughts and floods. Despite assumptions concerning "environmental migration" that posit immediate displacement following weather-related disasters such as floods, storms, and drought, the displacement process involves diverse responses and trajectories influenced by contextual conditions and individual capabilities rather than purely rational choices. In regions like Somalia, prone to recurrent extreme weather events, disasters and detrimental environmental conditions may trigger multiple instances of displacement, further amplifying vulnerability.

Environmental factors, progressively worsening as the climate becomes hotter and drier, directly impinge upon crop, livestock, and forestry-based livelihood systems. Regular and predictable rainfall is critical for the pastoral and rural communities across Somalia. Beyond the immediate effects of abrupt extreme weather events, the slow-onset of environmental degradation is a crucial factor provoking displacement towards more habitable regions. These rapid- and slow-onset environmental conditions may play divergent roles in individuals' decisions to migrate either internally or internationally at different times. Numerous qualitative and mixed-methods studies have shown that in many instances, the decision to migrate often materialises as a "last resort" in response to an extreme event, rendering existing living conditions unbearable [24–26]. To distinguish between new displacements and IDPs, Fig 1 denotes the number of movements, not individuals, as individuals can be displaced multiple times, a

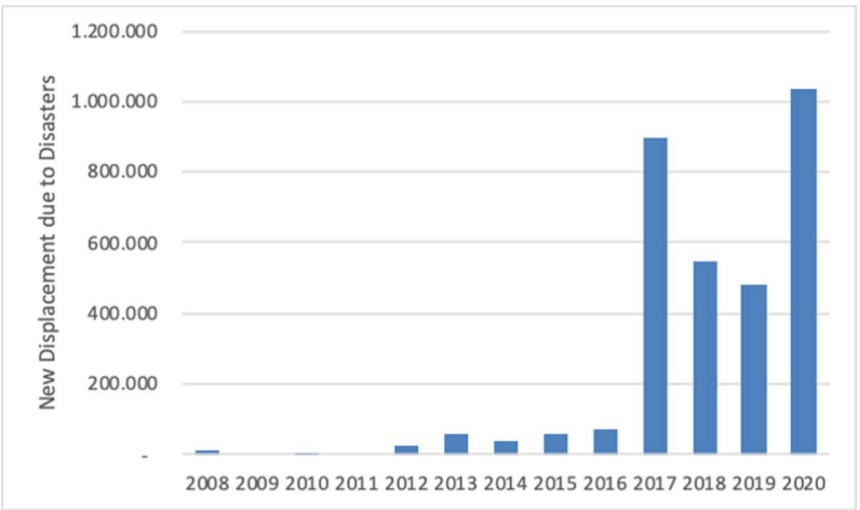

**Fig 1. Total number of new displacements due to disasters between 2008–2020 in Somalia.**

factor not always reflected in data collected by the Internal Displacement Monitoring Centre (IDMC).

Fig 2A presents the population density across Somalia, using a granular 100 × 100-meter grid square. Detailed views are shown around (a) Hargeisa, (b) Bosaso, and (c) Mogadishu. In Fig 2B, shows the distribution of two primary rivers in Somalia refers as breadbasket of Somalia with irrigated agriculture land producing Somalia food production. Shebelle, spanning the central areas, and Juba, crossing the southern regions of the country. The land cover of Somalia, as depicted by the European Space Agency (ESA) under the Africa land cover project in 2016, is represented in Fig 2C. Lastly, Fig 2D illustrates the proportion of population of each region displaced from their homes to other regions due to environmental factors. The IDP data was acquired from United Nations Human Rights Council (UNHRC). This map generated by aggregating the percentage of the population of each region displaced due to flood and drought between 2016 and 2019. These aggregate figures provide a macroscopic overview of Somalia's severe and erratic climatic conditions and the resultant human responses, often marked by difficult decisions to abandon their livelihoods. However, these annual figures inadequately elucidate the rapidly changing yet gradually evolving environmental dynamics and their indirect and direct impacts on human lives. Migration decisions can manifest both as an immediate reaction to disasters or a longer-term choice stemming from recurring environmental degradation and bleak long-term prospects. Given the diversity in individuals' responses to similar circumstances, dictated by their physical and economic limitations and their perception of short to long-term risks, it is challenging to discern the degree to which these movements contribute to the continuum of displacement. Thus, more time-sensitive environmental indicators can provide valuable insights into the relationship between climate alterations and internal displacement.

The scope of our study, delineated by the bold-blue line in Fig 2D, encompasses regions significantly affected by adverse climatic conditions, resulting in internal displacement. Criteria

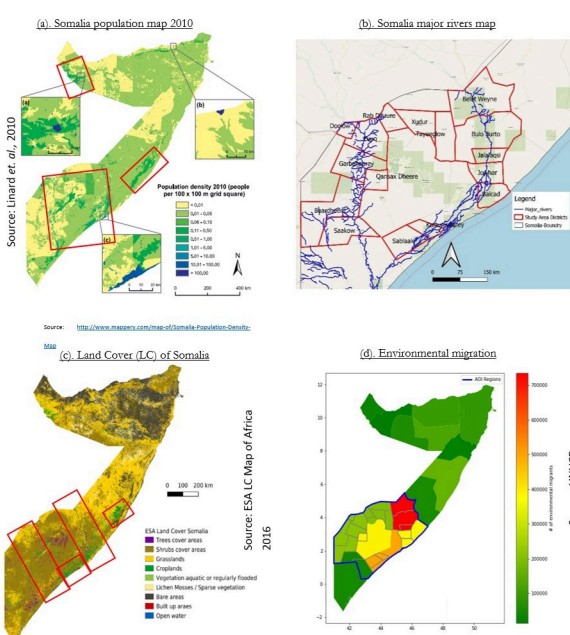

**Fig 2. Maps of Somalia.** a. The population density map across Somalia; b. Map of the major rivers in Somalia; c. Land-cover map of Somalia; d. The proportion of the population displaced across Somalia at regional level between 2016 and 2019. The extent coordinates for Somalia: WGS 84 (EPSG:4326) are 41, -1.7 to 51.4, 12.

for area selection involved considerations of population density, proximity to major rivers, and the degree of reported internal displacement attributable to environmental factors. Central and Southern Somalia were selected for this study, as these regions are not only densely populated (Fig 2A) but are also bisected by the Shebelle and Juba rivers (Fig 2B). This geographic configuration provides conducive conditions for crop cultivation and forage production (Fig 2C). Furthermore, these regions have consistently confronted internal displacement due to extreme environmental events (Fig 2D).

Fig 3 delineates the study area at the district level, with a total of 16 districts selected from the Central and Southern parts of Somalia, covering an area of 111,578 km$^2$. Detailed descriptions of the selected districts are provided in Table 1.

## Data

Our objective is to scrutinise the patterns of environmentally-induced internal displacement in Somalia by analysing the indicators we derived, aiming to discern any relationships with IDPs throughout the Area of Interest (AOI). We aim to understand the connection between extreme weather events, slow-onset climate change, and internal displacement in Somalia between 2016 and 2019. This study spans from January 1, 2016, to December 31, 2019. We chose the start year 2016 due to the availability of data from the Sentinel-1 (the first satellite in the Copernicus Programme launched by the European Space Agency) and Sentinel-2, an Earth observation mission from the Copernicus Programme that systematically acquires optical imagery at high spatial resolution over land and coastal waters, which we used to generate flood and drought maps. Furthermore, migration data has been available since 2015.

Previous studies that take a quantitative empirical approach to the climate-migration relation can be divided into micro studies, which largely focus on individual and household migration [26], and macro studies, where researchers examine environmentally induced migration at regional or national levels [9]. Regarding different hazard types, many studies focus on factors associated with climate change, such as changes in precipitation and temperature levels and variability [12]. This paper represents one of the first attempts to analyse climate/environment-induced displacement at the district level, employing environmental indicators derived from daily, weekly, and monthly remote sensing/satellite data.

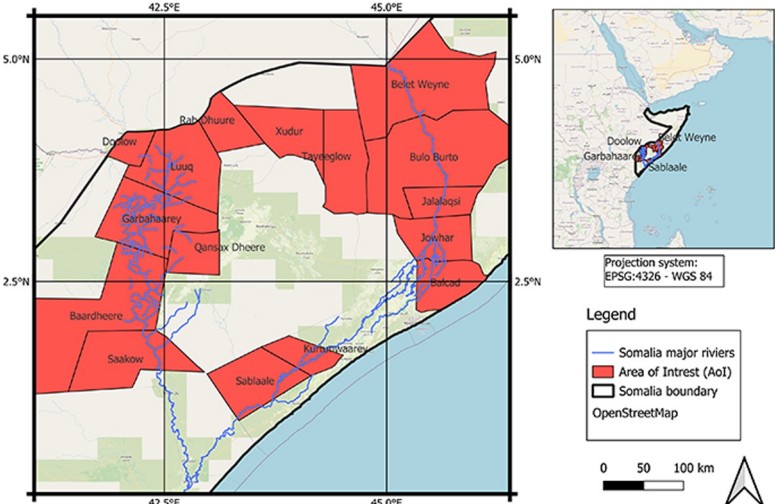

**Fig 3. Study area consist of sixteen districts with 111,578 km^2 coverage throughout centre and South Somalia.**

**Table 1. List of districts used as areas of interest (AOI) for agricultural drought indicator.**

| NO | Districts | District Area (km$^2$) | NO | Districts | District Area (km$^2$) |
|---|---|---|---|---|---|
| 1 | Saakow | 6774 | 9 | Bulo Burto | 16054 |
| 2 | Jowhar | 4637 | 10 | Baardheere | 15275 |
| 3 | Balcad | 4238 | 11 | Jalalaqsi | 3123 |
| 4 | Jalalqsi | 3123 | 12 | Kurtunwaarey | 2546 |
| 5 | Luuq | 8260 | 13 | Qansax Dheere | 3267 |
| 6 | Belet Weyne | 14867 | 14 | Rab Dhuure | 3309 |
| 7 | Bulo Burto | 16054 | 15 | Tayeeglow | 6596 |
| 8 | Garbahaarey | 8318 | 16 | Doolow | 1618 |

### Remote sensing data

**Sentinel-1.** Sentinel-1 offers an open-source Synthetic Aperture Radar (SAR) imaging satellite data with medium spatial resolution as part of the European Space Agency Copernicus programme. Sentinel-1 comprises of two satellites launched on 3 April 2014 and 22 April 2016. The SAR system operates within C-band (5.407 GHz) frequencies in one of four acquisition modes, with the default mode being Interferometric Wide Swath on land. In total, we downloaded and processed 32 SENTINEL-1 GRD scenes (16 pairs) from the Alaska Satellite Facility (ASF), spanning from 07/01/2016 to 30/10/2019. We used the Sentinel-1 satellite data to map the flood events in Somalia. Fig 4 shows Sentinel-1 acquired on 30th May 2016 during a flood event affected Beledweyne city. The areas near riverbank with black colour represent as the flooded zones where there are low backscatter. Furthermore, it has been observed that sandy areas located further from the river exhibit a similar low backscatter to the flooded areas and also represented in black.

**Sentinel-2.** The Sentinel-2 (Sentinel-2) dataset is an open-source optical with medium to high spatial resolution satellite data provided as a part of the ESA Copernicus programme. The S-2 mission consists of two satellites: Sentinel-2A and Sentinel-2B. Sentinel-2 provides multi-spectral data with 13 bands in the visible, near-infrared, and short-wave infrared parts of the spectrum. The S-2 is equipped with the state-of-the-art Multispectral Instrument, offering

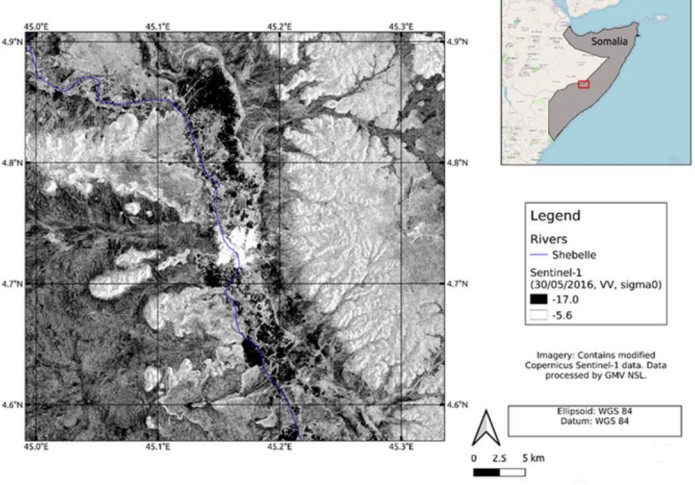

**Fig 4. Sentinel-1 imagery acquired over Belet Weyne city in Somalia on 30th May 2016.**

large swaths of high geometrical and spectral performance with imagery available at the spatial resolutions of 10 m, 20 m and 60 m. Sentinel-2 is broken down into 100 km x 100 km side tiles, known as "granules". In total, we selected 15 Sentinel-2 granules covering the 16 AOI districts. Fig 5 shows the distribution of Sentinel-2 granules over the study area.

On average, we downloaded 240 Sentinel-2 images for each granule from Google Cloud at the Level-1C processing level for a period of January 1st, 2016, to December 31[st], 201, covering an area of 180,757 km$^2$. In total, we downloaded 4169 of Sentinel-2 granules that occupied 2.5 TB of memory. We automatically pre-processed these datasets using Sen2cor (version 2.8) [27] to convert to Level-2A ready for processing. The Sentinel-2 datasets were used in several environmental indicators such as spectral vegetation indices, drought, flood, and Land Cover (LC) change. In addition, Sentinel-2 was used in flood mapping to visually verify the location and date of the flood events with an independent dataset.

**NASADEM.** The NASADEM is currently the Digital Elevation Model (DEM) with the highest resolution that is freely available for Somalia. NASADEM was derived from the re-processing of the Shuttle Radar Topography Mission, using the additional layers from Terra Advanced Spaceborne Thermal and Reflection Radiometer, Global Digital Elevation Model, Version 3 data, Ice, Cloud, and Land Elevation Satellite, Geoscience Laser Altimeter System ground control points of its LiDAR shots to improve surface elevation measurements that led to improved geolocation accuracy. We automatically downloaded NASADEM from the Land Processes Distributed Active Archive Center [28] in ".hgt" format distributed in 1˚ x 1˚ tiles (available for all land between 60˚ N and 56˚ S latitude), WGS84 for the year 2000 at approximately 30 m pixel spacing. A total of 45 granules were used to generate a DEM mosaic covering all the Sentinel-1 tiles. Other re-processing improvements include the conversion to geoid reference and the use of GDEMs and Advanced Land Observing Satellite Panchromatic Remote Sensing Instrument for Stereo Mapping AW3D30 DEM, and interpolation for void filling. The NASADEM was used to generate the following ancillary layers: a DEM mask, slope mask, and Height Above Nearest Drainage (HAND) index mask [24].

**Soil moisture index (SMI).** Soil moisture data represents the current state-of-the-art satellite-based soil moisture data record production, in line with the "Systematic observation requirements for satellite-based products for climate" defined by Global Climate Observing

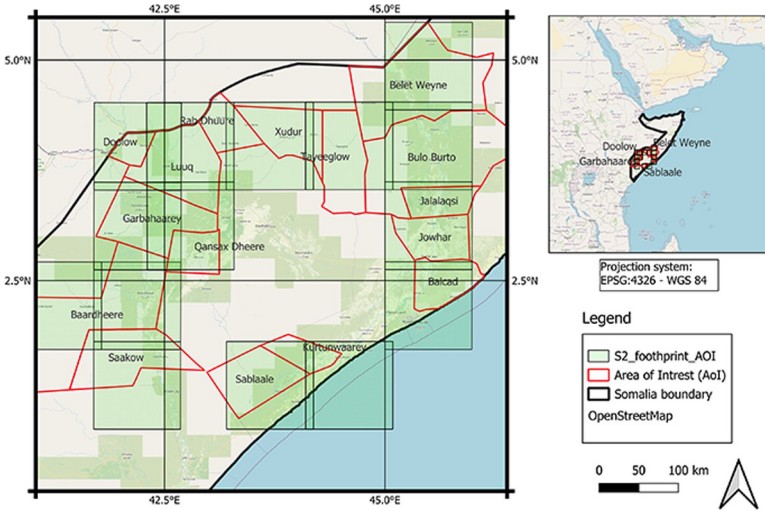

**Fig 5. The footprint of the Sentinel-2 granules acquired from January 2015 to December 209 covering the entire of the study area.**

System. The SMI offers a spatial resolution equal to 0.25-degrees and with a 10-day temporal resolution. We acquired SMI for the study area for the period between 1978 and 2019 at the ESA Climate Change Initiative Soil Moisture Climate Copernicus. The soil moisture data processed into SMI anomaly used to develop the agricultural drought indicator.

**Standardised precipitation-evapotranspiration index (SPEI).**   SPEI is a metric that delineates the climatic water balance, synthesising the variance between precipitation and potential evapotranspiration over diverse temporal scales [29]. Offering near real-time insights into hydric deficit or surplus, SPEI enables monitoring with a monthly cadence and 1-degree spatial resolution. While SPEI-1 delineates short-term, monthly meteorological anomalies, sensitive to abrupt climatic shifts such as sporadic rainfall events, Conversely, SPEI-3 encompasses a tri-monthly scope, yielding a more stable reflection of meteorological trends by mitigating the influence of transient climatic fluctuations. SPEI-1 and SPEI-3 data spanning from 1950 to 2019, sourced from the Spanish National Research Council [26], indicate SPEI's dynamic range from -5 (intense dryness) to 5 (significant wetness). Specifically, an SPEI value below -1 signifies a moderate drought, whilst a reading below -2 is indicative of an extremely drought scenario.

**IDP data.**   We compiled data from UNHCR, IDMC, and the World Bank where district-level statistics are available. The specifics of these derived and collected indicators are detailed in the ensuing section. As part of the non-environmental indicators, we used IDP data (sourced from UNHCR) to study the internal migration of environmental refugees since 2015. These IDP figures represent the number of internally displaced people each month and include the cause of migration and the originating and destination districts.

**Socio-economic indicators.**   Data availability and quality present significant challenges for socio-economic indicators in Africa, particularly in impoverished countries like Somalia [30–32]. This issue is further exacerbated when seeking disaggregated data for smaller administrative units. While international databases developed by various institutions cover diverse domains such as democracy, development, and access to services [33], they mostly provide annual statistics at the country or regional level, with limited availability of data for smaller administrative units.

To address this challenge, our first step involved selecting data sources that provided disaggregated socio-economic indicators at the district level on a monthly basis. We evaluated data from the World Bank, the International Organization for Migration (IOM), and the UNHCR to assess their quality and completeness for the study period from 2016 to 2019. Among these sources, we found that the most sufficient data is related to food prices at the district level on a monthly basis, providing information on the prices of essential commodities such as sorghum, maize, and rice.

It is essential to note that while we acknowledge the limitations, inconsistencies, incompleteness, and scarcity of data, the socio-economic variables were not selected as indicators for migration drivers. Instead, they were utilised to control for the impact of environmental factors on the socio-economic conditions in Somalia. By including these variables, we aimed to contextualise the influence of environmental factors on the broader socio-economic landscape of the region.

## Methodological approach: Developing environmental indicators

A series of environmental and non-environmental indicators were considered to understand environmental migration better. The goal of using environmental indicators was to measure the direct impact of environmental events on agriculture, human settlements, and public infrastructures, whilst non-environmental indicators such as IDP were used to quantify the human

impact. The environmental indicators were viewed as casual indicators, while non-environmental features measured the effect these causes had on the local population that ultimately leading to migration. Methodological approaches for each environmental indicator were developed and explained in detail.

## Agricultural drought indicator (ADI)

A methodology was developed for ADI, drawing on the cause-effect relationship for agricultural drought. This relationship posits that a shortage of precipitation leads to a soil moisture deficit, which in turn results in a reduction of vegetation productivity (see Fig 6). The methodology combines anomalies of precipitation (SPEI-1 and SPEI-3), soil moisture (e.g., SMI) and vegetation stress (e.g., NDVI). Here is how NDVI calculated in this study:

$$NDVI = \frac{NIR(band\ 8) - Red(band\ 4)}{NIR(band\ 8) + Red(band\ 4)} \tag{1}$$

- SPEI-1 and SPEI-3: to identify the precipitation deficit.

- SMI anomaly: to identify soil moisture anomalies.

- NDVI anomaly: characterise the subsequent effect of soil water stress on crops.

As illustrated in Table 2, a rainfall deficit (watch level) causes a reduction of the soil moisture (warning level), and a water deficit in the ground, leading to vegetation stress. This stress results in a reduction of NDVI values (alert level) which ultimately impacts agricultural production and could lead to food insecurity. The ADI developed for this study is a type of Combined Drought Indicator (CDI) derived by combining the SPEI, Soil Moisture Index anomaly (SMIa), and NDVI anomaly (NDVIa). SMIa is derived from anomalies of the SMI archive data

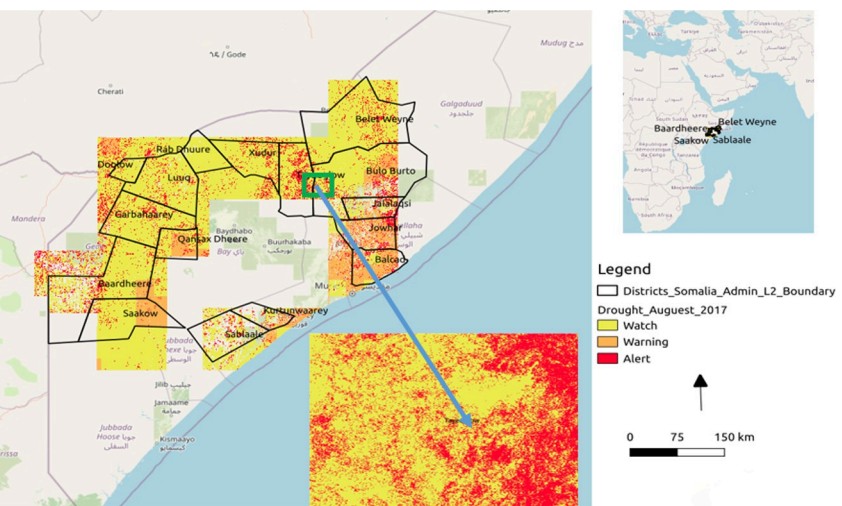

**Fig 6. Mosaic of Agricultural Drought (ADI) map of the study area in Somalia with 20m resolution in September 2017.** The extent coordinates for AOI: 41, 0.3 to 47, 5.4.

**Table 2. ADI classification scheme with three drought levels.**

| Drought level | Conditions | Interpretation |
|---|---|---|
| Watch (yellow) | SPEI-3 < -1 or SPEI-1 < -2 | Low SPEI means precipitation deficit. |
| Warning (orange) | SMI < -1 & (SPEI-3 < -1 or SPEI-1 < -2) | Precipitation deficit plus a negative anomaly of soil moisture. |
| Alert (red) | NDVIa < -1 & (SPEI-3 < -1 or SPEI-1 < -2) | Precipitation deficit plus a negative anomaly of vegetation 'greenness'. |

between 1992 and 2018.

$$\text{SMIa} = \frac{\text{SMI} - \text{SMI(avg)}}{\text{SMI(std)}} \tag{2}$$

SMI(avg) represents the long-term average soil moisture and SMI(std) corresponds to its standard deviation and SMI represents one soil moisture image corresponding to a specific date. NDVIa is an indicator of access vegetation health and productivity, and it represents deviations of the NDVI from the long-term mean values.

$$\text{NDVIa} = \frac{\text{NDVI} - \text{NDVI(avg)}}{\text{NDVI(std)}} \tag{3}$$

The expressions NDVI(avg) and NDVI(std) denote the long-term averages and standard deviation, respectively. Given the non-existence of the Sentinel-2 data prior to late 2015, the values for NDVI(avg) and NDVI(std) were derived from the Sentinel-2 archive spanning late 2015 to December 2019. It is imperative to note that the utilisation of Landsat data to augment the archive was not deemed viable. This decision was underpinned by the discrepancies in spatial and spectral band configurations between the Sentinel-2 and Landsat datasets, making them incompatible for cohesive analysis.

**Flood.**   The established methodology for flood mapping capitalises on the variation in backscatter intensity discerned from pre- and post-flood SENTINEL-1 SAR imagery. The SENTINEL-1 imagery was pre-processed using SeNtinel's Application Platform (SNAP) Toolbox 7.0 to carry out the following steps: (a) Apply orbit file, (b) Ground Range Detected (GRD) Border Noise Removal (border limit: 5000, trim threshold: 1.0), (c) Thermal Noise removal, (d) multi-looking (2 looks in range and 2 in azimuth), (e) speckle filtering (Refined Lee filter which is an enhancement of the Lee filter in order to provide better preservation of edges, linear features, and texture information, 5x5 window), (f) range Doppler terrain correction using SRTM 1arc sec and geo-coding to WGS84/UTM Zone 38N. The VV polarisation was considered to detect flooding as opposed to HH or VH polarisation [34, 35]. Intriguingly, full-polarisation SAR data extends the capability to apply advanced polarimetric techniques, such as 'target decomposition', which could potentially refine object detection further than raw polarised data alone. While beyond this paper's scope, future research could harness these techniques to heighten accuracy, especially in discerning specific objects within flood contexts.

Subsequently, pixel-based change detection between the pre-flood event at time ($t_1$) and the post-flood event at time ($t_2$) was performed using the intensity ratio:

$$R = \frac{I_{t1}(i)}{I_{t2}(i)} \tag{4}$$

Where "$I_{t1}(i)$" is the backscatter intensity of pixel "$i$" in image acquired at "$t_1$", and "$I_{t2}(i)$" is the corresponding intensity pixel in image acquired at time "$t_2$". The pixel ratio $R(i)$ is a sample of a random process whose Probability Density Function (PDF) for uncorrelated SAR

intensity images is [35]:

$$P(R) = \frac{\Gamma(2L)\gamma^L R^{L-1}}{\Gamma^2(L)(\gamma + R)^{2L}} \tag{5}$$

Where $\gamma$ is the true change of the radar cross sections $\sigma_0$, and $L$ is the number of looks. The probability of detection $P_{de}$ given a threshold T, L and $\gamma$ is:

$$P_{de}(R > T) = \int_T^\infty P(R)dR \tag{6}$$

It is important to notice that $P$ depends on the number of looks $L$. To illustrate the impact of $L$ on the detector performance let us compute the $P_{de}$ for the detection of a 3 dB change ($\gamma = 0.5$) with $T = 0.4$. For $L = 4$ (Sentinel-1) we have: $P_{de} = 0.58$, for $L = 50$ (Lee filter) $P_{de} = 0.86$. These figures point out the importance of speckle filtering for the ratio detector. In our implementation the threshold $T$ is determined by supervised learning, i.e., by estimating the ratio in a known area of no change, as it will be described in Thresholding section.

This approach minimises the misclassification of arid land as flooded due to similar backscatter of desert surfaces and flooded areas in arid regions like Somalia. Apart from common thresholding approaches were tested (e.g., unimodal thresholding approaches such as T-point and maximum deviation), a change detection approach was employed based on a ratio between pre-flood and post-flood imagery. This change detection strategy is designed to minimise the misclassification of arid landscapes as inundated terrain a frequent challenge due to the comparable backscatter characteristics of desert and water surfaces in arid zones such as Somalia. The current method's reliance on VV polarisation is rooted in established efficacy, yet it is recognised that incorporating target decomposition features, which utilise the comprehensive data from all polarisations, could substantially mitigate this issue. Such an advanced technique promises a refined analysis, leveraging the full spectrum of polarimetric data to distinguish between dry land and aquatic environments with greater precision. The thresholds in this study were calculated by supervised inspection of the imagery. The optimal threshold was calculated for a flooded pixel ($P_F$) and for a non-flooded pixel ($P_{NF}$). A coefficient $f_c$ was calculated based on several trial-and-error tests and also confirmed by findings in [36]. The $\mu$ and $\sigma$ are the mean and standard deviation of the no-change area selected in the ratio image $R$:

$$P_F \geq (\mu[R] + f_c * \sigma[R]) \, P_{NF} < (\mu[R] + f_c \sigma[R]) \tag{7}$$

Where $P_F$ is the flooded pixel; $P_{NF}$ is the non-flooded pixel; $f_c$ is a coefficient factor found based on several iterations; $\mu$ = mean $R$ and $\sigma$ = the $R$ standard deviation over the stable area.

Next, ancillary masks were generated based on NASADEM. A threshold was set to generate a binary slope mask. Slopes $\leq 3°$ were considered to be prone to flooding and assigned a pixel value of 1, while slopes $>3°$ are considered not likely to flood, assigned a pixel value of 0 and were removed from further analysis. This step was crucial in eliminating any pixels that may have exhibited a change in backscatter due to the angle of signal return from a higher elevation such as hills. Areas with an elevation $\geq 200$ m were also masked and deemed unlikely to be flooded. The slope and elevation threshold were selected based on the values used in the literature (e.g., [34, 37]).

Finally, a further terrain filter was applied to remove areas that can be considered unlikely to flood based on the Height Above Nearest Drainage (HAND) [38]. A binary HAND mask was generated where pixels with HAND values $\geq 15$ m were considered not flood-prone while pixels with HAND $< 15$ m were considered likely to be flooded. Post-processing of the resulting Sentinel-1 flood map was performed to remove isolated clusters of pixels $< 50$ pixels in

size (equivalent to 0.25 ha), reducing the false detections arising from noise. The algorithm removes and replaces pixels below the selected threshold with the pixel value of the largest neighbour polygon. Finally, a morphological binary closing was applied to fill holes smaller than the chosen disk radius ($r$ = 5) [39].

**Land Cover (LC) change.** A LC classification scheme was developed to study the impact of the natural disasters like drought and flood on agricultural food productivity and human activity. This motivated the inclusion of the cropland class, containing food producing arable land as well as food producing orchards. Meanwhile the Settlement class was created to include any apparent permanent human structure. The two principles "background" classes in this product where the Grassland/Shrubland class and the Soil/Sparse vegetation class. The primary differentiator between these two classes is the quantity of vegetation present, with Grassland/Shrubland dominated by low bushes, and occasional thick grass coverage, whilst the Soil class contains minimal vegetation. The study area is covered mostly with grasslands and shrubland whilst the majority of the croplands, forest, and settlement regions are distributed along the Shebelle River. The classification scheme can be seen below in Table 3:

The entire of four spectral bands of Sentinel-2 with 10m resolution (blue, green, red, near infrared) and two 20m resolution (Short Wave Infrared 1 and 2) were used at the Level-2A in the training and the evaluation of the model. Additionally, the NDVI was used to enhance the distinction between the cropland and forest classes.

A U-Net based model was chosen due to its superior performance in preliminary studies, both in terms of validation metrics and visual results. U-Net architecture is a type of Convolutional Neural Network (CNN) with layers that "slide" multiple trained filters to the image, creating several output channels. In U-Net, all convolutions have a 3x3 kernel size, meaning that a pixel and its immediate neighbours are all selected. The CNN implemented using Python 3 programming language.

The encoding part of the network used skip connections to form so called ResBlocks, which advantageously allow the input image to propagate deeper into this section of the network, thereby preventing issues such as vanishing gradients from being prominent. The loss function used in the network training was a weighted average of the cross-entropy and dice loss between the two images, with the dice cross comprising 90% of the loss function and the cross-entropy the remaining 10%.

The network was trained using the Adam Optimiser combined with a learning rate scheduler, which gradually reduces the learning rate throughout training by a constant factor of 0.92 per epoch. The network was trained for 100 epochs in total. The U-Net model had various hyperparameters which affected the details of the network architecture and training

Table 3. Classification scheme of AOI land cover with six members.

| ID | Class | Description |
|---|---|---|
| **0** | Agricultural Fields (green) | Arable land used for crop growing. Food producing orchards are also including in this class. |
| **1** | Forest (dark green) | Area covered by dense tree coverage, such that the underlying ground coverage cannot be seen. |
| **2** | Grassland / Shrubland (brown) | Open areas covered by dense non-arable vegetation including predominately from grass and bushes. Class may also include sparse tree coverage. |
| **3** | Settlement (grey) | Region forming a permanent human settlement. |
| **4** | Soil / Sparse vegetation (yellow) | Open area covered by bare soil, sand and low-density vegetation. |
| **5** | Water (blue) | Water bodies including lakes, rivers, ponds, and canals. Also includes regions of flooding covered by ground water. |

parameters that needed to be tuned in order to get an optimal result. There were two main hyperparameters that defined the structure of the U-Net, its depth, and its starting features. The depth of the U-Net is defined as the number of skip connections between convolutional encoding/decoding blocks. The depth of the model is 5, as there are 5 total skip connections between encoder/decoder blocks. The other key hyperparameter in the structure of the net-work was the number of starting channels that were used. A depth of 5 was shown to give the highest accuracy and F1 score on the validation set and was able to fit in the GPU memory. For the number of starting channels, as value of 64 was chosen by the same process as before of experimenting with values from 16 to 128, with 64 giving the highest accuracy on the validation.

For training and testing the U-Net model, a series of polygons were created using Sentinel-2 imagery. The polygons, selected during the two annual growing seasons, were devoid of any distortions. The U-Net model was trained on 128x128 images containing the seven input channels. The first approach of gathering training samples, inspired by an approach [40], syn-thetic training images were created from the training polygons. The algorithm to generate the synthetic training images via tessellation worked by randomly selecting a polygon from the training polygon set and then rasterizing it which involved calculating all input features. Each rasterised polygon was sampled randomly weighted by its area, meaning that larger polygons were more likely to be selected than smaller ones. In total, 1024 128x128 training images and 128 testing images of 128x128 were created.

The second approach to generating training images for the U-Net was to select a square ground region and attempt to label each pixel in the image using the Sentinel-2 data as a refer-ence. Each pixel in an image was hand labelled into one of the six classes defined in the classifi-cation schema. After augmentation this led to a total of 985 ground truth images of 128x128 in size, generation in total 2009 training images from both methods available for training. These polygons were then divided with 70% of the polygons being used for training and 30% for testing.

The LC Change was calculated based on the detection of changes between two LC maps: a before and after map. The LC change between the maps can then be calculated by performing a simple mapping of the form:

$$LCChange = LC_{after} + LC_{before} * \#\,classes \tag{8}$$

The number of classes was 6 resulting to a total of 36 possible LC Change classes represent-ing each possible transition of any class to any other class (including conservation). In the case where one or both LC values were unavailable for a given pixel due to the cloud mask, then the LC Change at this pixel was also masked. All possible transitions are seen in Table 4.

## Analysis and results

### Drought and floods

In total, 48 ADI mosaic were produced monthly with 20 m spatial resolution in GeoTIFF for-mat. Fig 7 shows an example of drought map from August 2017, where the AOI was predomi-nantly affected by moderate and severe drought, especially in the south near the lower Shebelle River.

River floods, caused by water overflowing riverbanks due to prolonged rainfall, are a main cause of environmental migration in Somalia. A flood algorithm was developed using a change detection technique with Sentinel-1 data from before and after a flood. The Digital Elevation Model was used to calculate the slope of the study area. Unlike the monthly drought maps, flood maps were produced based on flood events. Fig 8 displays the overall flood mosaic map.

**Table 4. Demonstration of the mapping of the transition between two land cover classes to an integer value that was stored in the land cover change raster file.**

| From \ To | Agricultural Field | Forest | Grass/ Shrub | Settlement | Soil | Water |
|---|---|---|---|---|---|---|
| **Agricultural Field** | 0 | 1 | 2 | 3 | 4 | 5 |
| **Forest** | 6 | 7 | 8 | 9 | 10 | 11 |
| **Grass/Shrub** | 12 | 13 | 14 | 15 | 16 | 17 |
| **Settlement** | 18 | 19 | 20 | 21 | 22 | 23 |
| **Soil** | 24 | 25 | 26 | 27 | 28 | 29 |
| **Water** | 30 | 31 | 32 | 33 | 34 | 35 |

In 2016, 865.5 km$^2$ of the AOI flooded, primarily affecting the Belet Weyne, Saakow and Jowhar districts (Fig 7A). In 2017, floods affected the districts of Saakow and Jowhar, covering 2263.4 km$^2$ (Fig 7B), whilst in 2018 several districts were flooded (including Hiiraan -Fig 6) with an area equal to 3552.1 $km^2$ (Fig 7C). Lastly, in 2019 the total area flooded was 2657.9 km$^2$. The flood was used as a base map alongside the population map to estimate the number of people affected by each flood event.

## Land Cover (LC) and LC change maps for drought and flood events

The final environmental indicator comprised of the LC and LC change maps, which are crucial for assessing the impact of natural phenomena like floods and drought on agricultural food productions (like orchards and croplands) and human settlements (like towns and villages). First, LC maps were produced to show the Earth's different land cover patterns in five different classes (soil/sparse vegetation, water, agriculture field, forest, and grassland/shrubland).

Sentinel-2 Level 2A imagery was the primary data input for LC maps. A cutting-edge U-Net based algorithm produced LC maps during drought and flood events, enhancing detection of different textural features seen in the base images. LC is a fundamental environmental variable for understanding the causes and trends of human and natural processes. LC refers to the physical material at the earth's surface like grass, soil, impervious surfaces, and trees. The information extracted from the land cover features and their associated changes can help

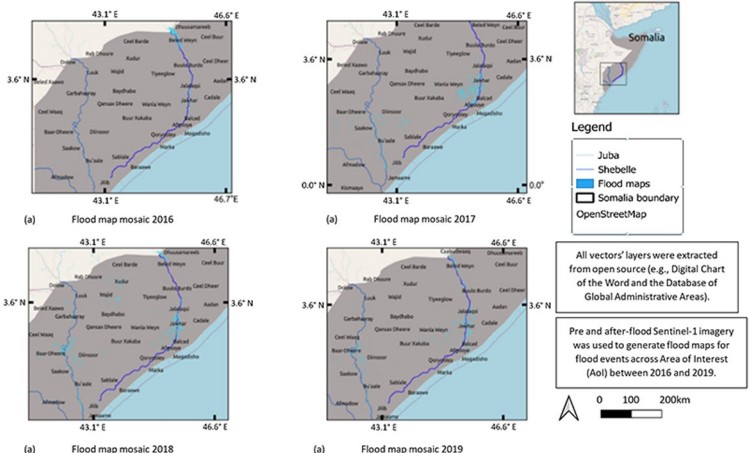

**Fig 7. Annual aggregated flood maps of the study area generated from Sentinel-1 data.** a. 2016 floods; b. 2017 floods; c.2018 floods; d.2019 floods.

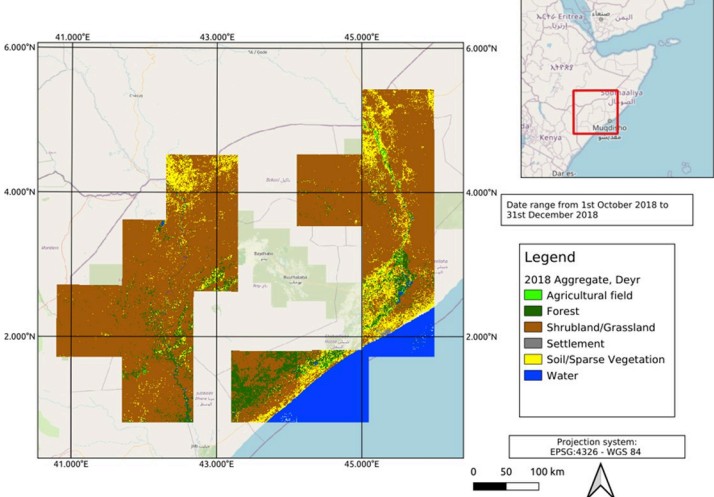

**Fig 8. Aggregated land cover map of the study area represent the Deyr growing season generated from Sentinel-2 time-series from 1$^{st}$ October to the end of the December 2018.**

understand the impact of environmental threats on the amount of variation of crops. Accurate identification and extraction of different land cover features (e.g., agricultural fields) is important for assessing distribution across agricultural regions. The Table 5 describe the statistical metric for the accuracy of the land cover model.

While the accuracy, precision, and recall on the test set provide an insight into the product's potential accuracy, a more rigorous accuracy assessment was carried out to obtain a closer estimate of the genuine accuracy. Random stratified points were chosen across the entire AOI, and very high-resolution imagery (e.g., Google Earth) was employed to determine the actual ground truth class for each point. The overall accuracy of the aggregated LC maps was estimated at 78%, a decline from the generalisation accuracy of 92% derived from the unseen test set. This decreased performance is likely attributable to the substantial amount of new, unseen data introduced to the model. Moreover, this dip in overall accuracy could be mitigated by referencing VHR imagery from Google Earth as ground truth, especially in instances where the Sentinel 2 imagery was indistinct or inconclusive.

In response, 17 LC maps were produced corresponding to significant flood events leading to IDPs from 2016 to 2019. The dates for creating LC maps were selected using the first available Sentinel-2 image after for each flood event. LC maps for drought events (see Fig 8) were generated as aggregates for growing seasons; Gu from March to late May and Deyr from the end of September to the end of December). Eight aggregate LC maps were produced, providing a general representation of Land Cover for each agricultural growing season between 2016

**Table 5. Precision, recall and F1 scores as calculated on a prediction of a mosaic comprised of single Sentinel-2 images from November 2017.**

| Class | Precision | Recall | F$_1$ Score | Support |
|---|---|---|---|---|
| **Agricultural Field** | 59.4% | 62.3% | 60.8% | 61 |
| **Forest** | 47.8% | 33.7% | 39.5% | 98 |
| **Shrubland/Grassland** | 60.3% | 80.7% | 69.0% | 243 |
| **Settlement** | 72.7% | 88.9% | 80.0% | 18 |
| **Soil/Sparse Vegetation** | 79.3% | 48.3% | 60.1% | 151 |
| **Water** | 94.5% | 92.9% | 93.7% | 56 |

and 2019. The Fig 8 shows the aggregate LC map on Deyr 2018. The Agricultural fields shows as light green colour represents the croplands and orchards mainly distributed across the She-belle and Juba rivers.

Following this, LC change maps were produced to highlight changes in agricultural fields and human settlements due to flood and drought events. LC change maps for flood events were generated by comparing post-flood LC maps to corresponding aggregate LC maps. Simi-larly, for drought, LC Change maps were produced by comparing each aggregate to the previ-ous years' aggregate map for the same season; for example, Gu 2016 was compared to Gu 2017.

Fig 9A presents the aggregated Land Cover (LC) map for Deyr 2019 over the city of Beled-weyne. A series of LC Changes were determined by comparing each LC map of the Gu and Deyr with the subsequent year's LC maps. These LC Change maps highlight the effects of drought and flood on settlements and agricultural fields. All the LC and LC change maps were produced with a 10m spatial resolution. The flood LC Change map in Fig 9B shows the clear impact of floods on the agricultural fields and settlements of the affected areas, whilst Fig 9C and 9D demonstrate the impact of the flood by showing the agricultural fields and settlements that were lost in the Beledweyne District, respectively. All the LC and LC change maps were produced in 10m spatial resolution. . . Most notably, evidence of migration from major urban areas close to the rivers to smaller communities far away from rivers can be seen after flood events. This suggests that migrants from major floods try to stay close to their homes but out-side the direct disaster zone.

An unsupervised clustering was performed to further explore the impact of flood events. Regular flood events exhibits similar characteristics, whereas extreme floods led to higher IDPs, more affected people, and caused larger damage to agricultural fields and human settle-ments. It is expected that, compared to regular floods that are part of the agricultural climate and essential for the growing season, extreme floods are more likely destructive and lead to migration.

Compared to flood events, drought events typically take place over a much longer time scale. As such for determining the LC change, a single post drought image cannot be defined in the same way as for flood. As a result, six drought LC change maps were produced based on

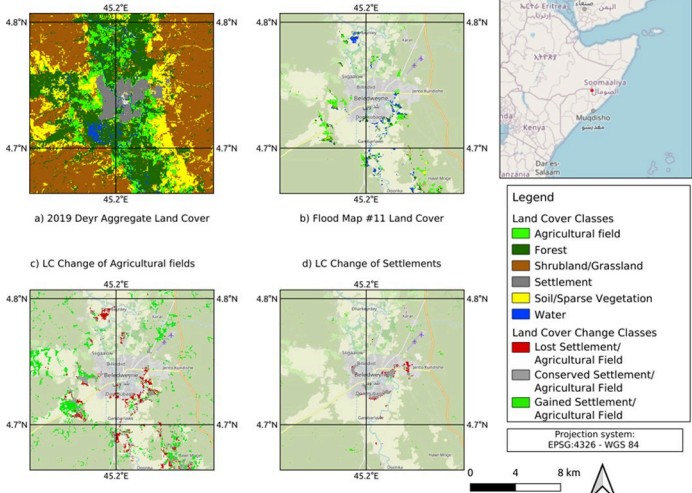

**Fig 9. Flood maps of the study area.** a. Aggregated LC map in Deyr 2019; b. LC Change map on flooded areas of Beledweyne after flood event (No 11); c. LC Change map with drought impact on agricultural fields; d. LC Change map with the impact of flood event (No 11) on settlements area.

comparing the matching seasonal aggregates LC. In addition, the change in the agricultural class was the only component of the drought LC Change, as drought should not directly destroy the settlement class as with flood. Fig 10 shows an example of the LC Change maps that were produced between the 2019 and 2018 Gu season showing the impact of a Drought in April 2019 around the Balcad region. Both Fig 10A and 10B depict the LC maps for April 2018 and 2019, respectively. Fig 10C displays the monthly drought map for April 2019. Taking into account the LC Change maps and drought maps, Fig 10D illustrates the impact of drought on agricultural fields. A significant portion of agricultural fields within drought regions (indicated in red) were either destroyed or damaged.

## Linking environmental factors and Internally Displaced People (IDPs)

To start with, Fig 11 presents a map of displacements taking place within and between district levels. During drought events from 2016 to 2019, Doolow emerged as the most preferred destination district for individuals displaced from their home districts. In total, nine migration movements were reported to Doolow, followed by six migration movements to Luuq. Conversely, Qansax Dheere had the highest number of departures due to drought events within the same period. Similar displacement patterns are showcased in Fig 12B. Notably, flood-induced displacements predominantly occurred within districts, suggesting that flood events were less likely to prompt individuals to leave their districts.

A series of plots were generated for all districts within our AOI to visually demonstrate the impact of environmental threats such as drought and flood on internal displacement in Somalia. We visually compared the plots of the Qansax Dheere and Doolow districts. The Qansax Dheere district had the highest outgoing environment-related displaced people, while Doolow district, on the border with Ethiopia, had the highest incoming environment-related displaced people.

**ADI and loss of agricultural fields.** In this study, agricultural fields (measured in hectares) are of significant interest because they provide food sources for humans and livestock and are the primary income source and means of survival in rural areas of Somalia. Fig 13 illustrates the impact of drought on variation of agricultural fields in the Qansax Dheere district from 2016 to 2019. The district's percentage affected by three different drought levels is

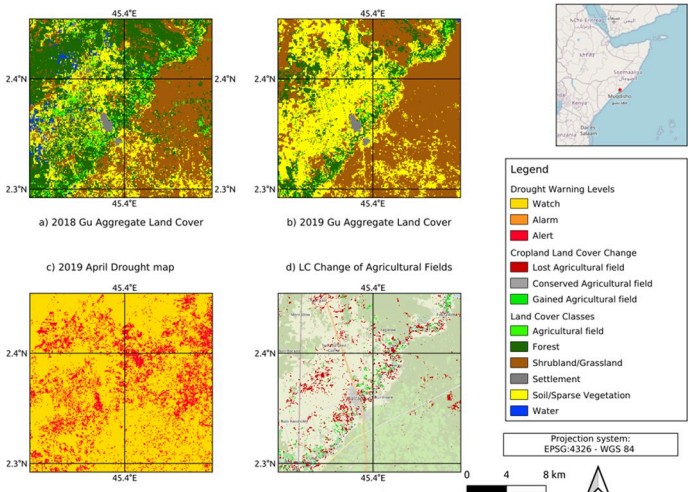

**Fig 10. Land cover maps.** a. Aggregated LC map in April 2018; b. Aggregated LC map in April 2019; c. ADI map of April 2019; d. LC Change map with the impact of drought on agricultural fields.

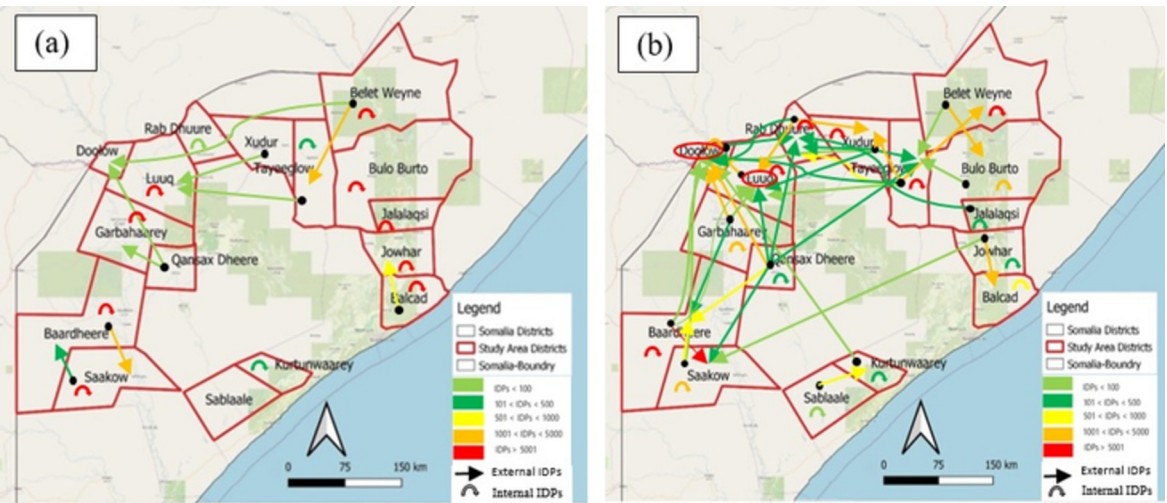

**Fig 11. Displacement maps.** a. Displacement movements occurred at the district level during drought event between 2016 and 2019; b. Displacement movements occurred at the district level during flood event between 2016 and 2019. The extent coordinates for AOI: 41, 0.3 to 47, 5.4.

represented by pale orange (watch level), orange (warning level), and dark brown (alert). The purple curve indicates the total agricultural fields lost.

As depicted in Fig 13A, the total agriculture fields started to decrease, shown by the upward movement of the purple curve, three months after the onset of drought in February 2016. In 2017, the agricultural fields began to disappear as a drought started in May, and despite an increase in drought intensity from May to July, the agricultural fields did not shrink. This minor reduction in the agricultural fields can be attributed to the natural increase in crop production that occurs during the agricultural growing season following the Gu growing season,

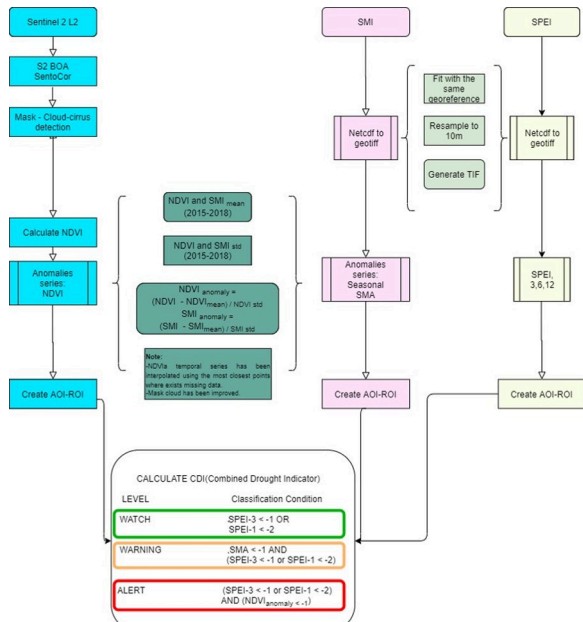

**Fig 12. Diagram of the developed methodology for the Agricultural Drought Indicator (ADI).**

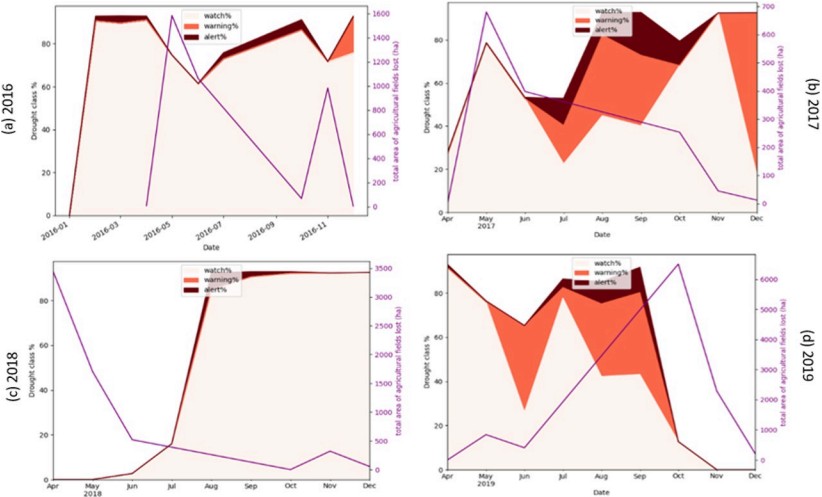

**Fig 13.** The impact of drought on agricultural fields in Qansax Dheere District during 2016 (a), 2017 (b), 2018 (c), and 2019 (d).

as seen in Fig 13B. Despite a greater proportion of the Qansax Dheere district being affected by the lowest degree of drought between August and December 2018, the reduction in the agricultural fields was mitigated. This can be explained by the significance of the drought degree impacting agricultural fields. In other words, agricultural fields are primarily affected by the most severe level of drought and least affected by the warning level of drought. This trend is observable in Fig 13D where coverage of agricultural fields began to diminish from June to October 2019 as more areas of the Qansax Dheere district were affected by alert and warning drought levels. Examination of plots from other districts tells a similar story: broadly speaking, agricultural fields decreased due to increasing drought severity and extent. These plots help validate our choice of the ADI metric by demonstrating a clear link to agricultural devastation, thereby supporting the proposed cause-effect hypothesis. It can be inferred that the ADI, particularly in severely affected regions, could indicate economic devastation, as seen in the reduction of agricultural fields.

**ADI and displacements.** A series of plots were generated for each of the sixteen districts to visualise the variation in IDPs during drought events. Here, we represent only two IDP plots for Qansax Dheere and Doolow. Qansax Dheere signifies the district with the highest outgoing migration events between 2016 and 2019. However, we selected the Doolow district due to its highest number of incoming displacements. Figs 14 and 15 provide a more comprehensive picture of displacement movements due to drought over the four-year study period, taking into account other factors including flood events and growing seasons. The background of each figure indicates the percentage of the district (Qansax Dheere, in the case of Fig 14) affected by watch, warning, and alert levels, as shown on the left vertical axis. Following this, two vertical strips per year represent the Gu and Deyr growing seasons, respectively. It is expected that precipitation will increase at the commencement of these seasons, thereby reducing the percentage and intensity of the drought through the growing seasons.

Three distinct IDP values are presented in the subsequent figures: net IDP, internal IDP, and total IDP. The net IDPs depicts the overall movement for a given month, considering both those leaving due to environmental factors and those migrating to this district to escape devastation from other districts. A negative net IDP indicates that more people moved into the district rather than departing. The internal IDP values reflect only the number of individuals

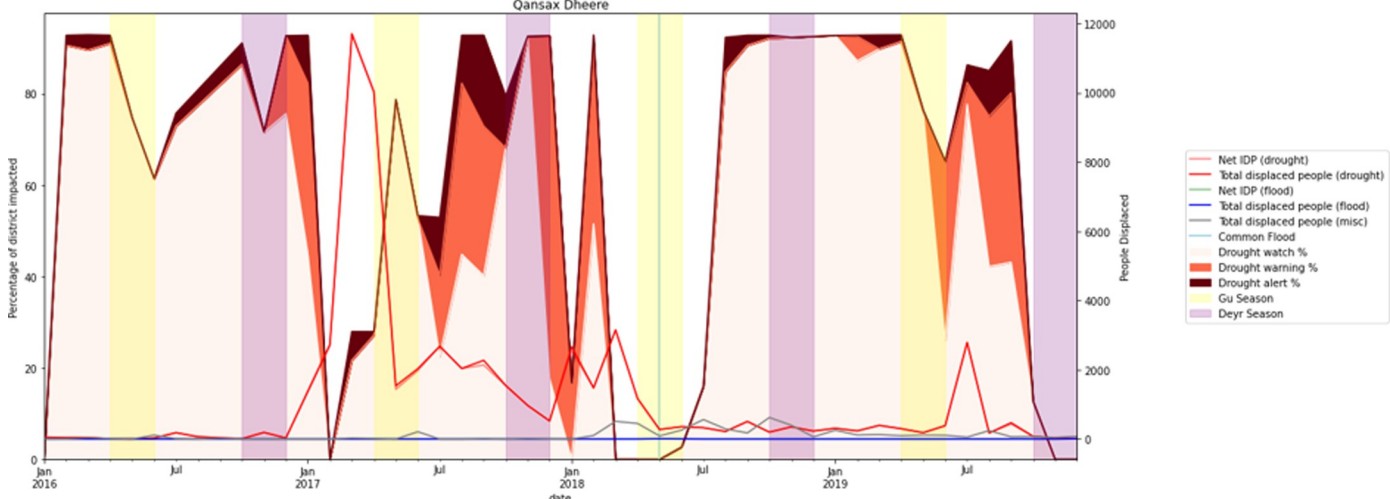

**Fig 14. Internal, total, and net IDP for the Qansax Dheere district compared to the Drought ADI classification and flood events for this same period.** Please note that the Gu and Deyr growing seasons are indicated on this figure as period of peak agricultural growth. Data for all indicators are generated on a monthly timescale.

displaced from their homes who opted to remain within their home districts, while the total IDP numbers present the total number of individuals displaced in some way over the course of a month.

As demonstrated in Fig 14, the initial displacement wave began two months after the onset of drought in November 2016 (the commencement of the dry season) and the majority of the displaced population returning to their homes in Qansax Dheere by May 2017. The remaining migrants took nearly a year to return to their homes. The second drought period spanned between June and September 2017, resulting in approximately 3,000 displaced people. This low number of displacements can be explained by the previous drought, during which many people had already left the district. Moreover, the drought that occurred during the agricultural growing season suggests there may have been enough food available for them to remain. The number of displaced people during the third drought period, from May to September 2019, did not exceed 3,000 people. This can be attributed to the availability of food during the agricultural growing season, much like the previous drought period.

Two major drought periods were identified for the Doolow district between 2016 and 2019 in Fig 15. The initial drought began in June 2016 and concluded in January 2017, resulting in a significant external displacement from Doolow two months later. The second significant drought event occurred a month after the first migrants had returned home, lasting until November 2018. As before, the migrants departed from Doolow two months later. This suggests that people might need to gather resources before they can move, and the decision to relocate can follow with a necessary time lag to realise the "capabilities" [9, 41].

While higher precipitation during the Gu and Deyr growing seasons might seem to decrease the impact of droughts decreases during these periods, we observed instances that contradict this expectation. In the case of Qansax Dheere, displacements were often external with no incoming migration. However, Doolow had a high negative net IDP, meaning that large numbers of people moved into Doolow due to drought in other districts is Somalia. This is not unexpected as people who managed to arrive in Doolow are closer to humanitarian support such as safety, food and healthcare provided at the nearby refugee reception centres and

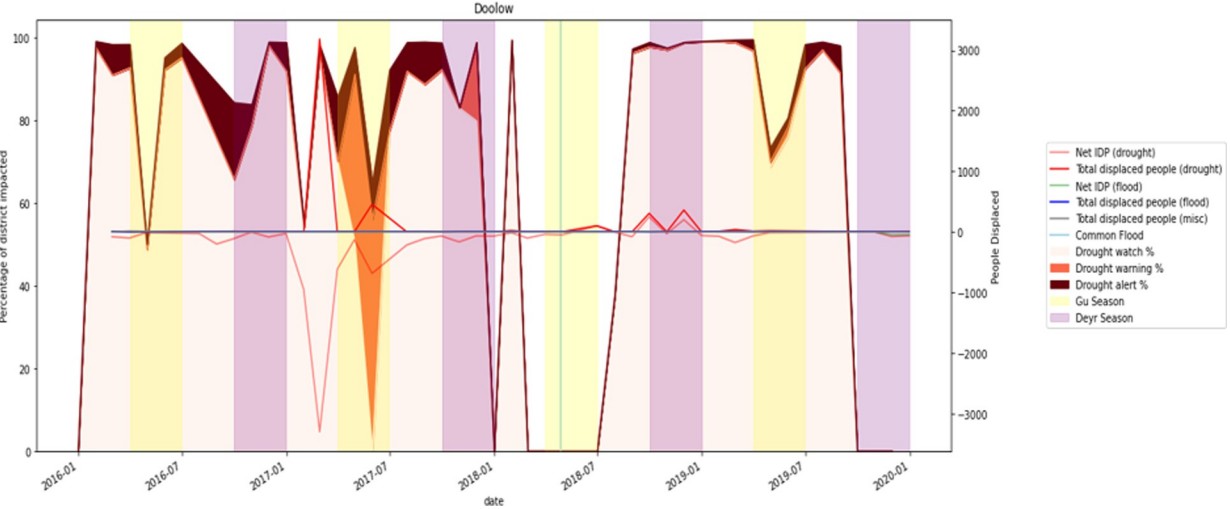

**Fig 15. The relationship between drought events and IDPs movement in Doolow district between 2016 and 2019.**

camps. Doolow is one of the primary exit points from Somalia to Dolo Ado, a small Ethiopian border town, and often serves as a hub for cross-border flows of displaced people.

Deyr

**Flood and displacements.** Compared to drought, the impact of flood events is largely localised and confined to specific districts. Fig 16 demonstrates that the majority of displacements due to floods remained internal within a district, with Belet Weyne recording two external displacements in addition to internal ones. The flood events representing regular and extreme floods, respectively, are depicted as two distinct lines. Of the three flood events, only two (April 2018 and November 2019) led to displacements movements in Belet Weyne district.

The effect of flood events on displacement is more immediate than that of drought. Internal displacement occurs shortly after the flood, followed by a swift return of the IDPs to their homes after the water recedes. In contrast, drought is more gradual, with people beginning to

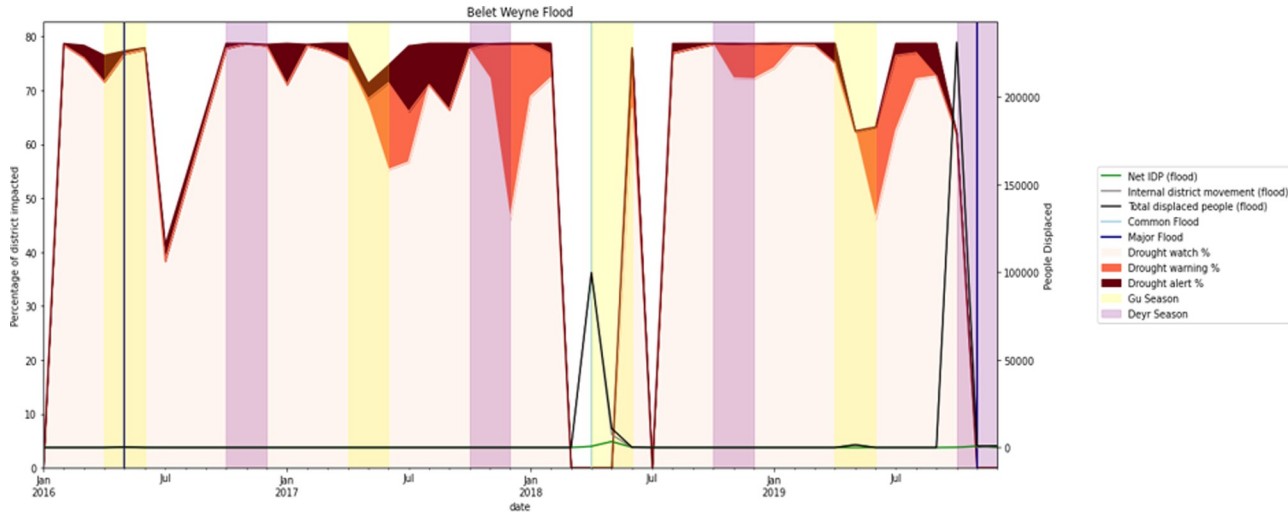

**Fig 16. The relationship between flood events and IDP movements in Belet Weyne district between 2016 and 2019.**

leave their homes with a delay after the drought starts. In addition, for those who opt to return home, it takes much longer, as it is likely that those migrating have decided that their previous living conditions were untenable.

In conclusion, the impact of droughts on departure and return displacements is more complex than that of floods. However, reviewing the entire AOI period on drought and IDPs for all districts reveals that a district affected by a severe drought does not necessarily result in a high displacement of people.

## Discussion and concluding remarks

In this study, we utilised remote sensing datasets to create various environmental indicators for Somalia including Agricultural Drought Index (ADI), flooding, and Land Cover (LC) changes. These were used to demonstrate the impact of extreme events on ground conditions in Somalia. Initially, the ADI methodology was developed around the cause-effect correlation found in agricultural droughts, where inadequate precipitation results in a soil moisture deficit, leading to decreased vegetation productivity. We generated a monthly time series of ADI, which corresponded to the different stages of the drought effects, categorised into three severity levels: watch, warning, and alert. Drought monitoring precision could be improved by supplementing data input with in-situ measurements of higher spatial resolution or by consistently using remote sensing and model ensemble techniques.

For flood mapping, a change detection and thresholding methodology was employed using, Sentinel-1 imagery closely timed to flood incidents. Historical satellite data before the Sentinel era could potentially augment our analysis; however, our methodological constraints, alongside the discernible disparities in spatial and spectral resolutions between Landsat and Sentinel series, deterred the integration of Landsat data. Despite the absence of Sentinel data pre-2015, our study navigated these limitations by focusing on the robust capabilities and temporal alignment of available Sentinel-1 and Sentinel-2 data post-launch, ensuring consistency in our remote sensing analysis. The absence of pre-2015 Sentinel data and lack of driect ground truth data for validating the ADI and LC change map constitute significant limitations, reinforcing the need for future studies to explore complementary datasets where feasible, while maintaining analytical coherence.

We developed a series of LC change maps to evaluate the repercussions of natural hazards, specifically drought and flood, on various land cover types, particularly those associated with agricultural productivity and human settlements. The LC methodology was constructed using the state-of-the-art U-Net technique, trained on the Sentinel-2 Level-2A dataset. The convolutional characteristic of U-Nets allows for the integration of information from surrounding pixels into predictions, offering superior detection of large-scale structures within images, compared to the single-pixel detection of Random Forests.

The U-Net training process was considerably expedited by collecting training data via polygons, circumventing the laborious task of manually labelling pixels across an entire image. As a result, the final trained model outperformed conventional machine learning techniques typically employed for land cover purposes, such as Random Forest. However, a notable limitation of LC change involves the inherent difficulty in disentangling the impact of drought from concurrent events throughout the entire growing season. These concurrent events could include flooding and deliberate shifts in cultivation patterns, potentially complicating the interpretation of the results. Our observations echo the findings of seminal works in this field [42–44], strengthening the validity of our assertions. Moreover, the nuances we've identified in the impact of concurrent events during the growing season, though briefly mentioned by [45, 46]

are expanded upon in our study. These interconnections, along with our unique observations, signify the broader relevance of our research and its positioning amidst existing literature.

Our analysis identified 2017 as the driest year for Somalia, with 2018 and 2019 being significantly impacted by floods. Regular rainfall periods, specifically the Gu and Dayer seasons, bring increased precipitation, which tends to mitigate drought impact during these growing seasons. In regions such as northern and central Somalia, where rapid-onset and slow-onset extreme climatic events occur either sequentially or concurrently, categorising environmental factors as beneficial or detrimental is intricate. This complexity is evident, for instance, when considering the mitigating effect of floods following severe droughts in agricultural areas. However, these environmental fluctuations may not always result in advantageous impacts on livelihoods.

As we delve deeper into how these extreme weather phenomena affect people's daily lives, it becomes clear that the consequences of rapid-onset events (like floods) and slow-onset events (like droughts) diverge, particularly in terms of individual or household response times. For instance, flood events prompt quick displacement, with minimal response time, indicating that people start evacuating shortly before floods occur, and often return to their homes soon after the floodwaters recede, contingent upon the area's recovery. Our study confirms that the effects of floods are predominantly localised and confined to districts, and most flood-induced displacement results in internal displacement within the same districts. Conversely, drought is a more progressive phenomenon, making adaptation to these more drastic changes challenging. Displacement due to drought begins with a delay and subsequently sees a gradual increase in the number of IDPs. Additionally, returning to habitual residences post-drought is a considerably extended process.

Our investigation into the underlying causes of displacement reveals a diverse and intersectional assortment of factors including climate-related threats, conflict, extreme poverty, and food insecurity. Given the profoundly inter-correlated nature of these factors and events, evaluating their marginal effects proves unfeasible. When individuals face the repercussions of these threatening conditions on their livelihoods, well-being, and security, whether in the short, medium, or long term, the decision to relocate can be construed as a strategic response. However, the predictability of events in terms of timing and the magnitude of their impact varies, leading individuals to opt for immediate reactions to avoid imminent starvation and material deprivation, rather than employing strategic responses.

Our research suggests that responses currently remain predominantly at an individual level, especially as mobility in response to slow-onset events significantly depends on individual "capabilities" [47]. Given projections of similar of greater magnitude climate change, developing a robust assessment of current and future impacts solely through macro analyses of aggregate socio-economic statistics of vulnerable regions and existing climate products seems unattainable. Specifically, detangling the intertwined aspects of economy, security, and climate appears to be an insurmountable challenge.

Climate change and extreme weather events not only trigger migratory movements but also exacerbate other factors such as economic deprivation and food insecurity. In the context of Somalia, a country grappling with both environmental hazards and armed conflict for decades, the urgent need for sustainable solutions to its humanitarian crisis is evident. Addressing internal displacement and migration in Somalia (and similarly afflicted countries) affectively will require resources beyond the Somalia's capacity and call for multinational and multilateral collaboration. The primary long-term strategy should aim at minimising the factors exacerbating climate change.

Considering climate-induced migration, understanding the intricate and multifaceted relationship between environmental variables and human migration goes beyond mere conceptual

discourse. A growing body of research highlights the inadequacy of treating climate factors as isolated and peripheral drivers of human mobility; a more nuanced comprehension is necessary to decipher the interactive dynamics intrinsic to this nexus (i.e., [9, 11]). Especially in a context like Somalia, which is susceptible to severe climatic conditions and persistent conflict, the fundamental causes of displacement prove challenging to explicate due to the relentless interconnections between conflict and environmental degradation.

Regarding the technical limitations of our study, the most significant hurdle was procuring a dataset - whether satellite or ground data - that aligns with the timescale and resolution of our AOI. For instance, to generate the flood product map, an image from Sentinel-1 immediately after the flood event is crucial to capture the flood's full extent. Nevertheless, the prolonged revisit time of the Sentinel-1 sensor sometimes resulted in unavailable images of the flooded region until several days after the flood began. This could lead to an inaccurate representation of the flood's full scope.

Moreover, for Land Cover products, the quality of ground truth data is crucial, as it directly affects the quality of the predictions. This includes a lack of ground truth data for validating past drought occurrences. While datasets like the Land Cover Net exist, they are often too expansive, encompassing the entire African continent, or they possess an unspecific classification scheme that doesn't match ours.

For other products, such as drought, our only alternative was to validate the predicted drought regions using reliable local drought reports. Future research on drought mapping should aim to improve the portrayal of drought status. This could be achieved either by enhancing data input through the use of in-situ measurements with finer spatial resolution, or by persisting with remote sensing and model ensemble techniques.

In addition, considering Synthetic Aperture Radar (SAR) data, full-polarisation techniques, particularly "target decomposition", can offer enhanced capabilities over raw polarised data such as VV, HH, and HV/VH when it comes to detecting specific objects. Although integrating such techniques was beyond the scope of the present study, it is indeed an avenue worth exploring. Future research endeavours with similar applications could greatly benefit from incorporating these advanced polarimetric techniques to achieve more accurate results.

Compounding the challenges of this study were the limitations we faced in accessing high-quality, micro, longitudinal socio-economic data, especially on a more granular level such as district-based data. The issue of data quality and availability remains a significant hurdle, especially for economically disadvantaged African nations [31, 32].

A key contributing factor in our case study of Somalia is the protracted armed conflict and war the country has been experiencing for decades. This has resulted in a dearth of governmental reports due to the inherent difficulties in collecting ground data, with only limited information available from international observers.

Despite these challenges, our study offers an innovative assessment of detailed environmental indicators and IDPs in Somalia. This approach could potentially be expanded to cover more extended time periods and wider geographical areas. Future research could examine cross-border mobility and environmental factors. However, given our findings and existing data challenges, we maintain that one of the first steps should be to improve data quality and availability at both the national and international levels for regions most affected by climate change.

Finally, we propose that triangulating quantitative and qualitative information can aid in illuminating such complex phenomena. This approach could underpin future research aimed at enhancing our understanding of climate-induced migration.

## Acknowledgments

We would like to express our sincere gratitude to our colleagues Haodong Qi and Miko Stanek for their critical review and insightful suggestions to improve our article.

## Author Contributions

**Conceptualization:** Tuba Bircan.

**Data curation:** Rahman Momeni, Tuba Bircan, Robert King.

**Formal analysis:** Rahman Momeni, Tuba Bircan, Eloy Zafra Santos.

**Funding acquisition:** Tuba Bircan.

**Methodology:** Rahman Momeni, Tuba Bircan.

**Project administration:** Tuba Bircan.

**Supervision:** Tuba Bircan.

**Visualization:** Rahman Momeni, Robert King, Eloy Zafra Santos.

**Writing – original draft:** Rahman Momeni, Tuba Bircan.

**Writing – review & editing:** Tuba Bircan.

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
