## [Decision Letter · Decision Letter 0]

27 Jun 2023

PONE-D-23-15881Deciphering Climate-induced Displacement in Somalia:

A Remote Sensing PerspectivePLOS ONE

Dear Dr. Bircan,

Thank you for submitting your manuscript to PLOS ONE. After careful consideration, we feel that it has merit but does not fully meet PLOS ONE’s publication criteria as it currently stands. Therefore, we invite you to submit a revised version of the manuscript that addresses the points raised during the review process. Please submit your revised manuscript by Aug 11 2023 11:59PM. If you will need more time than this to complete your revisions, please reply to this message or contact the journal office at plosone@plos.org. Please include the following items when submitting your revised manuscript:A rebuttal letter that responds to each point raised by the academic editor and reviewer(s). You should upload this letter as a separate file labeled 'Response to Reviewers'.A marked-up copy of your manuscript that highlights changes made to the original version. You should upload this as a separate file labeled 'Revised Manuscript with Track Changes'.An unmarked version of your revised paper without tracked changes. You should upload this as a separate file labeled 'Manuscript'.

We look forward to receiving your revised manuscript.

Kind regards,

Bijeesh Kozhikkodan Veettil

Academic Editor

PLOS ONE

Journal Requirements:

   "This research is supported by the European Commission through the Horizon2020 European project: “HumMingBird – Enhanced migration measures from a multidimensional perspective” (GA: 870661)"

   "MR, BT, KR

This research is supported by the European Commission through the Horizon2020 European project: “HumMingBird – Enhanced migration measures from a multidimensional perspective” (GA: 870661).

https://research-and-innovation.ec.europa.eu/funding/funding-opportunities/funding-programmes-and-open-calls/horizon-europe_en

4. Please amend the manuscript submission data (via Edit Submission) to include author Robert King.

5. Please amend your authorship list in your manuscript file to include author Richard King.

6. We note that Figures 2-6 and 8-12 in your submission contain [map/satellite] images which may be copyrighted. All PLOS content is published under the Creative Commons Attribution License (CC BY 4.0), which means that the manuscript, images, and Supporting Information files will be freely available online, and any third party is permitted to access, download, copy, distribute, and use these materials in any way, even commercially, with proper attribution. For these reasons, we cannot publish previously copyrighted maps or satellite images created using proprietary data, such as Google software (Google Maps, Street View, and Earth). For more information, see our copyright guidelines: http://journals.plos.org/plosone/s/licenses-and-copyright.

a. You may seek permission from the original copyright holder of Figures 2-6 and 8-12 to publish the content specifically under the CC BY 4.0 license.  

7. Please include a separate caption for each figure in your manuscript.

Reviewers' comments:

Reviewer's Responses to Questions

**Comments to the Author**

1. Is the manuscript technically sound, and do the data support the conclusions?

Reviewer #1: Partly

Reviewer #2: Yes

2. Has the statistical analysis been performed appropriately and rigorously? 

Reviewer #1: No

Reviewer #2: No

3. Have the authors made all data underlying the findings in their manuscript fully available?

Reviewer #1: No

Reviewer #2: No

4. Is the manuscript presented in an intelligible fashion and written in standard English?

Reviewer #1: No

Reviewer #2: Yes

5. Review Comments to the Author

Reviewer #1: The work has a good theme but needs changes:

* Needs a grammatical review, giving special attention to words, such as line 5 (acknowledgement), line 14 (emphasise), line 25 (neighbouring), and so on.

* The text has several statements, which do not contemplate the results of the work, without a theoretical basis (citations). Bringing more citations in the text is of paramount importance.

* The research addresses climate change, but its study period is from 2016 to 2019, which under no circumstances represents climate change. Other satellites, in addition to the sentinels, could be used for at least another 25 years, with migration data available from 2015 onwards. The union of these data would contribute to the SMI and SPEI data.

* It remains to elucidate in the introduction how remote sensing can contribute to studying the relationship between the climatic indicator and the displacement of people.

* The formulas in lines 327, 333, 338, 356, and 447 do not contain references.

* Lines 472, 519, 539, 554, 562, 568, 587, 591, and 603 also do not contain references.

* In the discussions, you also need some references to confirm the n statements made (ex: line 689). As well as to represent an honest discussion and not just a summary of the work.

* The maps in Figures 2,6, and 8 need geolocalized coordinates.

* All figures are without description.

Reviewer #2: In this study the optical (S-2) and SAR (S-1) images were used along with some other environmental data such as soil moisture, precipitation-evapotranspiration, and digital elevation model to develop climate indicators which were utilized together with some socio-economic indicators to study the relationship between climate factors and internally displaced persons in a district level in Somalia.

A relatively high amount of technical work has been done, which makes the paper technically acceptable. However, there are still shortcomings such as the lack of quantitative assessment of the accuracy of the classification results. Please find the attached file to see all my comments and suggestions.

6. PLOS authors have the option to publish the peer review history of their article (what does this mean?). If published, this will include your full peer review and any attached files.

Reviewer #1: No

Reviewer #2: No

---

## [Author Response · Author response to Decision Letter 0]

14 Feb 2024

Deciphering Climate-induced Displacement in Somalia: A Remote Sensing Perspective

Response to the Reviewers

We would like to thank reviewers for their constructive comments. Please see our response elaborated below for each comment.

The revised manuscript was controlled for PLOS ONE’s style requirements and updated accordingly.

 "This research is supported by the European Commission through the Horizon2020 European project: “HumMingBird – Enhanced migration measures from a multidimensional perspective” (GA: 870661)"

 "MR, BT, KR

This research is supported by the European Commission through the Horizon2020 European project: “HumMingBird – Enhanced migration measures from a multidimensional perspective” (GA: 870661).

https://research-and-innovation.ec.europa.eu/funding/funding-opportunities/funding-programmes-and-open-calls/horizon-europe_en

The amended statements are included in the new cover letter.

The data availability) for this specific paper depends on the HumMingBird project (which is the main funding channel for the proposed work) data availability planning. There is a limitation of maximum 52 MB of data can be uploaded into CESSDA platform. However, because of the minimum memory size of each data exceeds 700 MB, the initial products of processed raw data cannot be publicly available. Nevertheless, the aggregated and anonymised data used for this paper is being prepared to be publicly available at Consortium of European Social Science Data Archives (CESSDA) data catalogue and Harvard Dataverse repository as of the end of the HumMingBird project (June 2024).

4. Please amend the manuscript submission data (via Edit Submission) to include author Robert King.

The authorship list amended for Robert King. We also added another co-author (Eloy Zafra Santos), as agreed by all existing co-authors.

5. Please amend your authorship list in your manuscript file to include author Richard King.

The authorship list amended for Robert King. We also added another co-author (Eloy Zafra Santos), as agreed by all existing co-authors.

6. We note that Figures 2-6 and 8-12 in your submission contain [map/satellite] images which may be copyrighted. All PLOS content is published under the Creative Commons Attribution License (CC BY 4.0), which means that the manuscript, images, and Supporting Information files will be freely available online, and any third party is permitted to access, download, copy, distribute, and use these materials in any way, even commercially, with proper attribution. For these reasons, we cannot publish previously copyrighted maps or satellite images created using proprietary data, such as Google software (Google Maps, Street View, and Earth). For more information, see our copyright guidelines: http://journals.plos.org/plosone/s/licenses-and-copyright.

a. You may seek permission from the original copyright holder of Figures 2-6 and 8-12 to publish the content specifically under the CC BY 4.0 license. 

All figures are recreated and all logos that might create a copyright issues were removed from the figures.

7. Please include a separate caption for each figure in your manuscript.

Separate captions and relevant legends are included within the text for all figures.

Reviewers' comments:

Reviewer's Responses to Questions

Comments to the Author

1. Is the manuscript technically sound, and do the data support the conclusions?

Reviewer #1: Partly

Reviewer #2: Yes

2. Has the statistical analysis been performed appropriately and rigorously? 

Reviewer #1: No

Reviewer #2: No

3. Have the authors made all data underlying the findings in their manuscript fully available?

Reviewer #1: No

Reviewer #2: No

4. Is the manuscript presented in an intelligible fashion and written in standard English?

Reviewer #1: No

Reviewer #2: Yes

5. Review Comments to the Author

Reviewer #1: The work has a good theme but needs changes:

* Needs a grammatical review, giving special attention to words, such as line 5 (acknowledgement), line 14 (emphasise), line 25 (neighbouring), and so on.

Thanks for pointing out the need for language improvement. The revised text does not only address content related comments, but it was also edited to ensure high quality language use in writing. Accordingly, the revised version has gone through a thorough language editing (for British English) to improve the language and the narrative. All abbreviations were used only after the full names/text were introduced first. Difficult to understand sentences are rewritten. Besides, the full text was proofread by a native speaker to avoid typos and grammatical mistakes. 

* The text has several statements, which do not contemplate the results of the work, without a theoretical basis (citations). Bringing more citations in the text is of paramount importance.

The text is revised and further and more recent scientific citations are added.

* The research addresses climate change, but its study period is from 2016 to 2019, which under no circumstances represents climate change. Other satellites, in addition to the sentinels, could be used for at least another 25 years, with migration data available from 2015 onwards. The union of these data would contribute to the SMI and SPEI data.

Thank you for your insightful remark on the study period and the potential use of other satellites to extend our temporal scope.

While the study period of 2016 to 2019 may seem limited in the context of climate change, our focus was on utilising the Sentinel series for its unique attributes and high-resolution data. Our methodology involved the development of unique indicators from scratch, rather than employing pre-existing products or indicators. This choice, although presenting certain advantages in the precision and relevance of our findings, introduced significant complexities.

Merging diverse data sources from satellites other than the Sentinels becomes especially challenging when creating custom indicators, as opposed to using off-the-shelf products. Different satellites often possess varying spatial and spectral resolutions, calibration systems, and other sensor-specific traits. Integrating these diverse datasets for a seamless and scientifically robust output required meticulous calibration, correction, and harmonisation processes, which, given our specific approach, could have introduced inconsistencies and uncertainties.

Furthermore, the processing of such vast and diverse datasets is not only technically challenging but also immensely time-consuming. Our indicators, being unique, required tailored pre-processing and analysis steps, each of which demands substantial computational resources and time. Extending our study period to incorporate another 25 years of data would have multiplied the processing time and could have jeopardised the timely completion of the study.

That said, we recognise the value in a more extended temporal analysis to capture the nuances of climate change more comprehensively. We are keen to delve deeper into this in future projects, possibly by collaborating with experts in satellite data harmonisation, or by employing more advanced computational techniques and resources.

* It remains to elucidate in the introduction how remote sensing can contribute to studying the relationship between the climatic indicator and the displacement of people.

The introduction is revised and we added a new paragraph (p.1) to demonstrate the significance of remote sensing to investigate the climate indicators and human displacement. 

* The formulas in lines 327, 333, 338, 356, and 447 do not contain references.

The message “Error! Reference source not found.” Appear where anchoring to the referred figures and maps failed to appear when the submitted Word file is converted to pdf by the submission system. The anchors are removed to avoid such error messages. Formulas are also updated and revised accordingly.

* Lines 472, 519, 539, 554, 562, 568, 587, 591, and 603 also do not contain references.

The message “Error! Reference source not found.” Appear where anchoring to the referred figures and maps failed to appear when the submitted Word file is converted to pdf by the submission system. The anchors are removed to avoid such error messages.

* In the discussions, you also need some references to confirm the n statements made (ex: line 689). As well as to represent an honest discussion and not just a summary of the work.

Thank you for pointing out the oversight in our discussion section. We understand the importance of substantiating our statements with credible references to lend more weight and credibility to our assertions. In response to your comment:

We thoroughly reviewed the discussion section, especially at the mentioned line (line 689), to identify instances where claims or statements can be strengthened with appropriate references from the existing body of knowledge.

We also acknowledge that the discussion section should not merely be a summary of the work but should delve deeper into the implications, comparisons, and relevance of our findings in the broader context of the field. To address this, we integrated new text (p.35) with scientific references from the field on more comparative and critical analyses, drawing parallels and distinctions with established research in the field, and pr

---

## [Decision Letter · Decision Letter 1]

27 Mar 2024

PONE-D-23-15881R1Deciphering Climate-induced Displacement in Somalia: A Remote Sensing PerspectivePLOS ONE

Dear Dr. Bircan,

Thank you for submitting your manuscript to PLOS ONE. After careful consideration, we feel that it has merit but does not fully meet PLOS ONE’s publication criteria as it currently stands. Therefore, we invite you to submit a revised version of the manuscript that addresses the points raised during the review process.

We look forward to receiving your revised manuscript.

Kind regards,

Bijeesh Kozhikkodan Veettil

Academic Editor

PLOS ONE

Journal Requirements:

*Additional Journal Comments: Please see the attached document for some additional minor revisions/clarifications requested by the reviewer.*

Reviewers' comments:

Reviewer's Responses to Questions

**Comments to the Author**

1. If the authors have adequately addressed your comments raised in a previous round of review and you feel that this manuscript is now acceptable for publication, you may indicate that here to bypass the “Comments to the Author” section, enter your conflict of interest statement in the “Confidential to Editor” section, and submit your "Accept" recommendation.

Reviewer #1: (No Response)

Reviewer #2: All comments have been addressed

2. Is the manuscript technically sound, and do the data support the conclusions?

Reviewer #1: Yes

Reviewer #2: Yes

3. Has the statistical analysis been performed appropriately and rigorously? 

Reviewer #1: Yes

Reviewer #2: Yes

4. Have the authors made all data underlying the findings in their manuscript fully available?

Reviewer #1: Yes

Reviewer #2: No

5. Is the manuscript presented in an intelligible fashion and written in standard English?

Reviewer #1: Yes

Reviewer #2: Yes

6. Review Comments to the Author

Reviewer #1: (No Response)

Reviewer #2: Thanks to the authors. Most, but not all, of the proposed comments have been addressed. The manuscript has been improved and can be accepted.

7. PLOS authors have the option to publish the peer review history of their article (what does this mean?). If published, this will include your full peer review and any attached files.

Reviewer #1: No

Reviewer #2: No

---

## [Author Response · Author response to Decision Letter 1]

7 May 2024

All minor remarks are addressed including language edits, extra clarifications within the text as well as the bibliography formatting.

---

## [Editor Report · Decision Letter 2]

8 May 2024

Deciphering Climate-induced Displacement in Somalia: A Remote Sensing Perspective

PONE-D-23-15881R2

Dear Dr. Bircan,

We’re pleased to inform you that your manuscript has been judged scientifically suitable for publication and will be formally accepted for publication once it meets all outstanding technical requirements.

Kind regards,

Bijeesh Kozhikkodan Veettil

Academic Editor

PLOS ONE
---

## [Editor Report · Acceptance letter]

15 May 2024

PONE-D-23-15881R2 

PLOS ONE

Dear Dr. Bircan, 

I'm pleased to inform you that your manuscript has been deemed suitable for publication in PLOS ONE. Congratulations! Your manuscript is now being handed over to our production team.

Kind regards, 

on behalf of

Dr. Bijeesh Kozhikkodan Veettil 

Academic Editor

PLOS ONE